# Saliva and plasma metabolome analysis during the five days before foaling in the mare

**Lydie Nadal-Desbarats[1,2], Camille Dupuy[1,2], Frédéric Montigny[2], Priscila Silvana Bertevello[1], Cécile Douet[3], Amandine Gesbert[4], Fabrice Reigner[4], Ghylène Goudet ⓘ[3]***

**1** Unité Mixte de Recherche 1253, iBrain, Institut national de la santé et de la recherche médicale (INSERM), Université de Tours, Tours, France, **2** Plateforme de Métabolomique et d'Analyses Chimiques, US-61 ASB, Université de Tours, CHRU Tours, Institut national de la santé et de la recherche médicale (INSERM), Tours, France, **3** Unité Mixte de Recherche de Physiologie de la Reproduction et des Comportements (PRC), Institut National de Recherche pour l'Agriculture, l'Alimentation et l'Environnement (INRAE), CNRS, Université de Tours, Nouzilly, France, **4** Unité Expérimentale de Physiologie Animale de l'Orfrasière (PAO), Institut National de Recherche pour l'Agriculture, l'Alimentation et l'Environnement (INRAE), Nouzilly, France

* ghylene.goudet@inrae.fr

## Abstract

Saliva is a relevant biofluid for real-time welfare-friendly monitoring of systemic events in animals, because some bioanalytes have a systemic origin and its collection is painless, stress-free and non-invasive. Our aim was to analyze the metabolome of equine saliva during the five days before foaling, with a focus on identifying metabolites whose level significantly changed before parturition, that could be potential salivary biomarkers of the onset of parturition. We compared the saliva and plasma metabolomes to investigate their relationship. Saliva and blood samples were collected from twelve mares once a day in the morning, from 322 days of gestation to the day of foaling. Samples collected from four days before the day of parturition (D-4) to the day when parturition occurred (D0) were analyzed by [1]H Nuclear Magnetic Resonance spectroscopy. We identified 50 metabolites in saliva and 51 in plasma. In saliva, the levels of three metabolites and three groups of metabolites showed significant differences between the days. In particular, acetic acid significantly decreased three days before D0 and again on D0, isovaleric acid significantly increased from four days before D0 to D0, and lactic acid significantly decreased between three and two days before D0. In plasma, the levels of D-Glucose and four groups of metabolites showed significant differences between the days. D-Glucose significantly increased between three and two days before D0, and again on D0. In conclusion, significant changes in the salivary metabolome have been shown in the antepartum period in the mare. However, only minor changes in the levels of these metabolites were observed, without any significant threshold that would allow the prediction of foaling. Significant modifications of the plasma level of glucose have been observed before foaling. The development of non-invasive glucose monitoring

**Data availability statement:** The experimental data have been deposited with the DOI https://doi.org/10.5281/zenodo.17589754 in the Zenodo repository (https://zenodo.org).

**Funding:** This work was supported by Institut National de Recherche pour l'Agriculture, l'Alimentation et l'Environnement (INRAE) and Institut Français du Cheval et de l'Equitation (IFCE). The funders had no role in study design, data collection and analysis, decision to publish, or preparation of the manuscript.

**Competing interests:** The authors have declared that no competing interests exist.

sensors could allow the development of non-invasive detection method for the prediction of foaling.

## Introduction

In most of the foaling in domestic mares, no human intervention is necessary. However, when problems arise, they may lead to the death of the foal or the mare, associated with ethical, emotional and financial concerns. It is therefore desirable to attend parturition, for early intervention if necessary or rapid veterinary assistance. However, the duration of gestation is highly variable in mares, ranging from 306 to 365 days, and foaling often occurs at uncertain times, late at night or early morning [1,2].

Traditional detection of mare parturition relies on direct observation, which is time-consuming and labor-intensive, and can introduce a stress to the mare. Moreover, physical and behavioral signs of readiness for parturition are variable among mares and may be subtle, and observation of clinical signs such as relaxation of the pelvic ligaments, development of mammary gland, waxing of the teats, or increased vulvar length are highly variable, inconsistent, and not precise enough to predict parturition [2–4]. Accelerometers attached to the halter or the tail base are currently used to detect mare parturition, with significant changes detected during the last hour before foaling [5,6]. However, behavior changes are variable between mares and are biased by housing, management systems and environmental stressors. Algorithms for automatic visual detection of mare parturition behavior using video images are currently under development [7]. However, they still have some limitations, particularly regarding their applicability across different environments, breeds and conditions, and the demand for technological equipment and expertise. Skin temperature was also proposed to detect mare parturition [8]. However, the rise in skin temperature in the last 90 minutes prepartum was only 0.5°C and temperature sensors are easily influenced by environmental factors such as room temperature, water and dust.

Biomarkers of parturition have been searched in mammary gland secretions. Changes in consistency, color, pH and concentration of total protein, albumin, lactose, immunoglobulins, chloride, sodium, potassium and calcium in mammary gland secretions occur before foaling [3,4]. A decrease in pH and an increase in calcium concentration are used for prediction of parturition, however they predict that foaling is imminent in only 80% of mares, they are most reliable to predict when the mare is not ready to foal and cut off values are different among studies [3,4]. A reduction in conductivity of mammary gland secretions, related to pH and electrolyte concentrations, is associated with impending foaling [9]. However, repeated attempts to collect mammary secretions may be stressful, painful and may damage the mammary gland or make it more sensitive, impairing the acceptance of the foal suckling and the future lactation.

Major changes occur in the plasma of the pregnant mare close to parturition. During the last three days before foaling, cortisol concentration in maternal plasma increases markedly, concomitant with a decrease in pregnenolone and progestogens

including 5alpha-dihydroprogesterone and its hydroxylated metabolites [4,10,11]. The level of five microRNAs in the plasma of pregnant mares showed significant changes between 30 days before foaling and two days before foaling [12]. These molecules with changing levels close to parturition could potentially be used as biomarkers to predict foaling. However, repeated plasma collection is invasive, potentially painful and stressful. Moreover, gas or liquid chromatography tandem mass spectrometry that is necessary for specific and reliable measurement of steroid hormones [13], and extraction of microRNAs, sequencing, reverse transcription quantitative PCR and bioinformatic analyses are expensive and time-consuming techniques that may not allow stall-side tests with results promptly available for breeders.

Thus, detection of mare parturition is still challenging, and achieving real-time, accurate, painless, non-invasive and stress-free detection is of significant importance for ensuring the safety of mares and foals, improving ethical aspects and economic benefits for horse farms.

Among biological fluids, saliva gains interest since it can be collected by non-invasive, painless, stress-free and easy sampling methods, allowing the collection of repeated samples. Saliva is composed of secretions from the minor and major salivary glands and molecules from the plasma. The systemic origin of some of its bioanalytes make saliva a pertinent fluid to determine systemic events and physiological status. Analysis of saliva composition using omics approaches such as metabolomics have been used to identify salivary biomarkers of the physiological status in horses [14,15]. Omics approaches are powerful tools to identify large numbers of biomolecules in biological fluids. The metabolome includes low-molecular-weight compounds such as lipids, amino acids, carbohydrates, organic acids, peptides and vitamins, that are end-products of intracellular biochemical reactions organized in pathways whose activity is modified during physiological changes, such as hormone-induced modifications during the reproductive cycle [15]. The metabolome is thus the signature of a physiological state. Moreover, many commercial colorimetric assay kits are available for easy, quick and cheap analysis of metabolites level, which could allow to develop inexpensive and fast assays for diagnostic tests in the field.

Thus, our aim was to analyze the metabolome of equine saliva in the last five days before foaling, with a focus on identifying metabolites whose level significantly changed just before parturition, that could be potential salivary biomarkers of the onset of parturition in a welfare-friendly production system. We compared the saliva and plasma metabolomes to investigate the relationship between these two biofluids according to the physiological stage.

## Materials and methods

### Animals, housing, saliva and blood samples collection

Ethics statement: This study was carried out in strict accordance with the recommendations in the Guide for the Care and Use of Laboratory Animals of the French Ministry of Agriculture. The protocol was approved by the Committee on the Ethics of Animal Experiments n°019 of the French Ministry of Research (Comité d'Ethique en Expérimentation Animale Val de Loire n°019) under number APAFIS #34147−202111261451233 v2. This study follows the Animal Research Reporting of In Vivo Experiments (ARRIVE) guidelines. This experiment follows the three Rs principles: Replace, Reduce, Refine. Replacement: animals were used because no suitable alternative method exists. Reduction: the number of animals was minimized to expose as few animals to experimental conditions as possible, and the statistical analysis was appropriately designed to obtain reliable results. Refinement: housing and management of animals and experimental conditions were optimized to respect animal welfare and alleviate suffering and stress, as detailed below. No sacrifice, anesthesia or analgesia was required in this experiment.

Twelve healthy Welsh-type pony mares with singleton pregnancies, aged 6–13 years old, were reared in our experimental farm (Unité Expérimentale de Physiologie Animale de l'Orfrasière, UEPAO, Animal Physiology Facility, https://doi.org/10.15454/1.5573896321728955E12) from INRAE (Institut National de Recherche pour l'Agriculture, l'Alimentation et l'Environnement) in France. From December to March, mares lived in groups on straw bedding in an indoor stall under natural daylight, with free access to an outdoor paddock during the day. They were fed with 0.6 kg of concentrate (20% oats, 18% wheat straw, 16% wheat bran, 15% barley, 12% alfalfa; Eperon INRA, Axereal, Saint-Germain de Salles,

France), with salt licks (Sodical, Levalois-Perret, France), hay and water available ad libitum. From April to foaling in May and June, mares lived in groups in the pasture with salt licks and water available ad libitum. Parturition occurred in the field, no complications occurred and all foals were healthy. After foaling, mares and foals lived in groups in the pasture with salt licks and water available ad libitum.

From 322 days of gestation to the day of parturition, saliva and blood samples were collected every morning between 8 AM and 9 AM to avoid circadian variations of steroids secretions. The day when foaling occurred was called D0, saliva and blood were collected in the morning and foaling occurred in the night for all the mares, so that the timing between sampling and foaling was less than 24 hours.

To avoid saliva contamination with food, mares were brought in boxes without straw with water available ad libitum one hour before saliva and blood sampling. Saliva was collected first, using a cotton swab (Sarstedt Salivette® ref. 51.1534.500; Sarstedt, Nümbrecht, Germany), held with forceps in the mouth of the mare following an animal-welfare respectful procedure. The mare was allowed to chew on it until it was soaked. The cotton swab was centrifuged at 3000×g for 5 min and the recovered saliva was stored at −80°C until analysis. After saliva collection, blood samples were collected from the jugular vein with a 5mL Vacutainer® heparinized tube and a 20G needle following an animal-welfare respectful procedure. Blood samples were centrifuged at 4000×g for 10 min and the recovered plasma was stored at −80°C until analysis. Saliva and blood collection were well tolerated by the mares without any restraint needed. After samples collection, the mares went back to the pasture.

Saliva and plasma samples collected four days before the day of parturition (D-4), three days before the day of parturition (D-3), two days before the day of parturition (D-2), one day before the day of parturition (D-1) and the day when parturition occurred (D0) were analyzed.

## Metabolome analysis by ¹H Nuclear Magnetic Resonance (NMR) spectroscopy

The saliva and plasma samples were prepared as previously described (https://doi.org/10.1016/j.theriogenology.2024.08.007 and https://doi.org/10.3390/metabo11100681). The 60 saliva samples were prepared by protein precipitation using cold methanol. Briefly 200 µL of saliva were mixed with 400 µL of cold methanol, vortexed, centrifuged at 15,000 g at 4°C during 15 min. Then, 500 µL of supernatant were collected and evaporated in a SpeedVac (Thermo Fisher Scientific, Illkirch, France) before storage at −20°C until NMR analysis. The 60 plasma were prepared by a modified Folch's method. Briefly 200 µL of plasma were mixed with 300 µL of cold methanol and 500 µL of cold chloroform, vortexed, centrifuged at 15,000 g at 4°C during 15 min. Then, 300 µL of supernatant were collected and evaporated in a SpeedVac (Thermo Fisher Scientific, Illkirch, France) before storage at −20°C until NMR analysis. Ten quality control samples were prepared with a pool of 20 µL of each sample (either saliva or plasma) and were further prepared with the other biological samples.

Before NMR analysis, 210 µL of deuterated phosphate buffer (pH = 7.4; containing 3-trimethylsilylpropionic acid (TSP) at 152 µM final concentration) were added to the dried residues and transferred to 3 mm NMR tubes.

¹H-NMR spectra were acquired at 298 K on an AVANCE III HD 600 MHz system (Bruker Biospin, Karlsruhe, Germany) equipped with a 5 mm TCI cryoprobe with z-gradient. ¹H-NMR spectra were recorded with a "noesypr1d" pulse sequence with a time domain of 64 K data points, a sweep width of 12 ppm, 64 scans, an acquisition time of 4.56s and a relaxation delay of 20s. A NUS (50%) TOCSY sequence was used on the quality controls for the metabolite identifications of the biological samples.

The ¹H-NMR spectra were phase corrected and imported in NMRProcFlow (https://doi.org/10.1007/s11306-017-1178-y) for baseline correction and bucketing. Regions with a coefficient of variation across the quality control inferior to 30% were kept for the analysis.

The metabolite identification in the ¹H-NMR spectra were done using the NMR 2D TOCSY information and ChenomX NMR suite 8.8 (ChenomX Inc, Edmonton Canada).

 

## Metabolite set enrichment analysis

As previously described [15] (https://doi.org/10.1016/j.theriogenology.2024.08.007), to identify the most represented chemical classes in the NMR detected metabolomes of the saliva and plasma, a metabolite set enrichment analysis was performed on the metabolites identified in each sample type using Metaboanalyst (MetaboAnalyst 6.0, http://www.metabo-analyst.ca). The metabolite enrichment analysis of each set of metabolites was performed using over-representation analysis (ORA) with the 'sub chemical class metabolite sets' provided, containing 1250 sub chemical class metabolite sets. Enrichment is calculated by comparing the number of identified metabolites observed in a sample (hits) for a chemical class compared to the total number of metabolites expected in that class (expected). The p-value were corrected for multiple comparisons by the false discovery rate (FDR) method.

## Statistical analysis

Two-way ANOVA followed by Fisher's multiple comparisons test was performed using GraphPad Prism version 10.5.0 for Windows, GraphPad Software, Boston, Massachusetts USA, www.graphpad.com. A $p\text{-value} < 0.05$ was considered significant.

## Results

The experimental data have been deposited with the DOI https://doi.org/10.5281/zenodo.17589754 in the Zenodo repository (https://zenodo.org).

The $^1$H NMR spectra of the saliva extracts allowed the identification of 125 spectral regions or buckets, corresponding to 50 known metabolites (Fig 1A). The $^1$H NMR spectra of the plasma allowed the identification of 78 spectral regions or buckets, corresponding to 51 known metabolites (Fig 1B). S1 Table resumes all the identifications of the metabolites in saliva and plasma with their chemical shifts in ppm, their multiplicities and their corresponding identifiers according to the Metabolomics Standards Initiative (MSI; https://doi.org/10.1007/s11306-007-0082-2). There is a discrepancy between the number of buckets and metabolites due to the fact that each moiety of a metabolite gives a signal leading to more signal regions than the number of metabolites in a biological sample. Another concern is the overlapping of different signal coming from different compounds leading to a bucket accounting for a group of metabolites. Among buckets representing only one metabolite, we identified 25 metabolites in both saliva and plasma, 25 only in saliva and 26 only in plasma (Fig 2).

In order to identify the over represented chemical class metabolite set in saliva and plasma metabolome, a metabolite set enrichment analysis (MSEA) was performed. Fourteen chemical classes were over-expressed in both saliva and plasma with a $p\text{-value} < 0.05$ (Fig 3). The list of metabolites in each common chemical class between saliva and plasma is presented in S2 Table. This table resumes the number of observed metabolites (hits) in each chemical class. The three chemical classes having the higher number of hits in its metabolite set are amino acids, peptides and analogues, carboxylic acids, fatty acids and conjugates. Eighteen amino-acids and analogues (*Creatine*; Dimethylglycine; *gamma-Aminobutyric acid; Glycine; L-Tyrosine; Phenylalanine; L-Alanine*; L-Threonine; L-Asparagine; *Isoleucine;* Histidine; Serine; Creatinine; Glutamine; *Leucine;* Methionine; *L-Valine; Phosphocreatine*) enriched the amino-acids and analogue set in the plasma metabolome (p-value = $6.2\text{E}^{-20}$) while 14 (Betaine; *Creatine; gamma-Aminobutyric acid; Glycine; L-Tyrosine; Phenylalanine; L-Alanine; Isoleucine*; Ornithine; L-Arginine; *Leucine; L-Valine; Phosphocreatine*; 5-Aminopentanoic acid) enriched the saliva one (*p*-value = $3.27\text{E}^{-14}$). Ten compounds (*italic in the text*) are common to the two biofluids. The carboxylic acid set were enriched with Acetic acid; Formic acid; Propionic acid; Isobutyric acid (*p*-value = $2.84\text{E}^{-10}$) in plasma and with Acetic acid; Formic acid; Propionic acid (*p*-value = $1.26\text{E}^{-7}$) in saliva. The fatty acid set is enriched with five compounds, 2-Hydroxy-3-methylbutanoic acid, Capric acid, 3-Hydroxyisovaleric acid, Methylsuccinic acid, 2-Hydroxyvaleric acid in plasma ((*p*-value = $4.1\text{E}^{-6}$) and Butyric acid, 2-Hydroxy-3-methylbutanoic acid, Isocaproic acid, Isovaleric acid, Valeric acid in saliva (*p*-value = $3.7\text{E}^{-6}$).

In saliva, the levels of six metabolites or groups of metabolites with overlapping signals were significantly different between the days before parturition (Fig 4). The intensities (in arbitrary units) of the bucket containing Lactic acid and

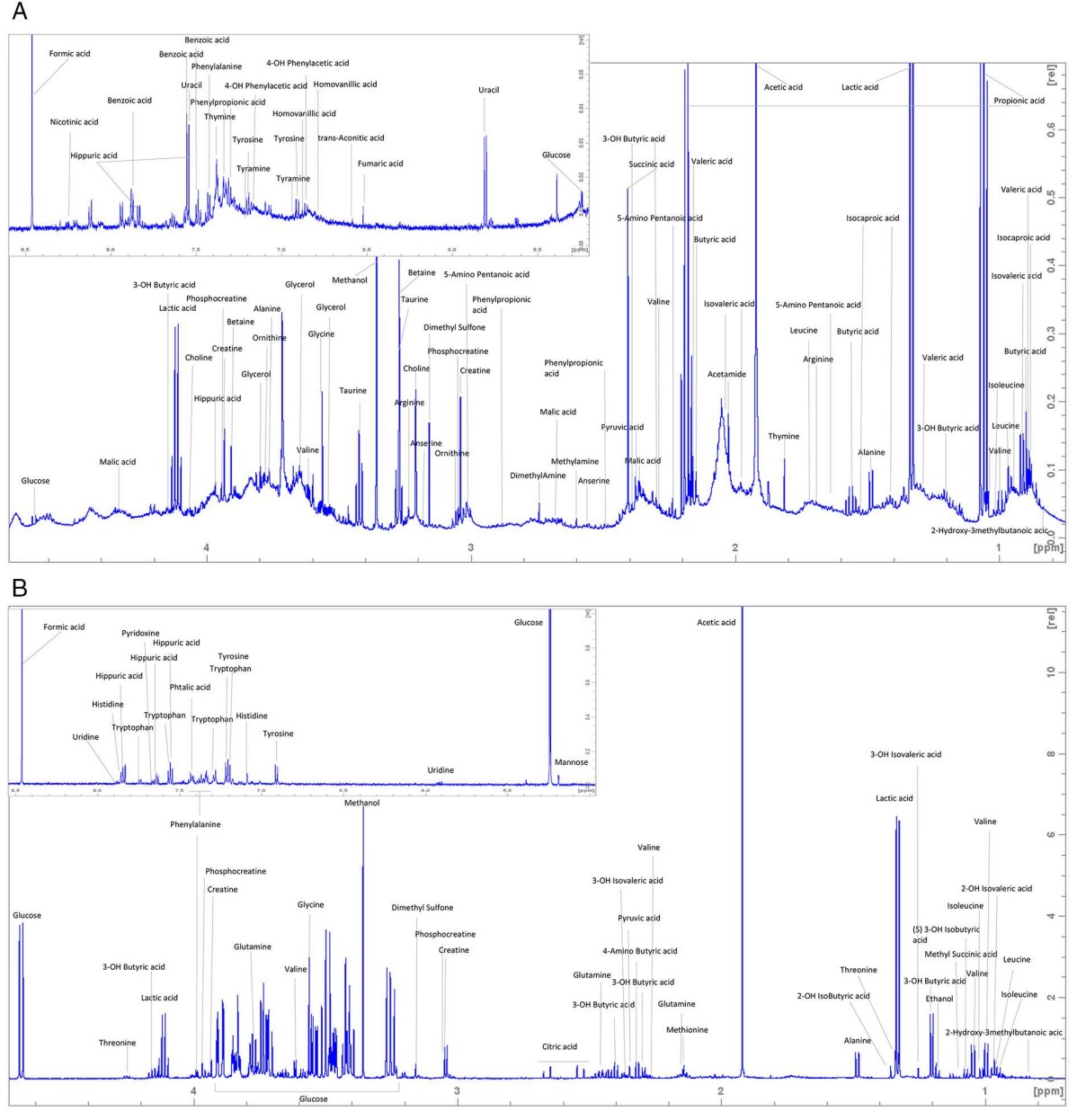

**Fig 1. Annotated representative ¹H-Nuclear Magnetic Resonance spectroscopy spectra of mare saliva (A) and plasma (B).**

Valeric acid significantly decreased two days before the day of parturition (D-2), and significantly increased on the day when parturition occurred (D0). The salivary levels of Acetic acid significantly decreased three days before the day of foaling (D-3) and significantly decreased again on the day of foaling (D0). The salivary levels of Isovaleric acid significantly increased from four days before the day of parturition (D-4) to the day of foaling (D0). The intensities of the group of metabolites containing Methanol and Phenylalanine significantly decreased between two (D-2) and one (D-1) day before the day of parturition, but with high individual variations. The salivary levels of Lactic acid significantly decreased between three (D-3) and two (D-2) days before the day of foaling, and then increased but non-significantly and with high

### Metabolites found in both compartments

| | | |
|---|---|---|
| Acetamide | Glycerol | Leucine |
| Acetic acid | Glycine | Methanol |
| L-Alanine | Hippuric acid | Phenylalanine |
| gamma-Aminobutyric acid | 3-Hydroxybutyric acid | Phosphocreatine |
| Creatine | 2-Hydroxy-3-methylbutanoic acid | Propionic acid |
| Dimethylamine | Isoleucine | Pyruvic acid |
| Dimethyl sulfone | Ketoleucine | L-Tyrosine |
| Formic acid | Lactic acid | L-Valine |
| D-Glucose | | |

### Specific saliva metabolites

| | |
|---|---|
| trans-Aconitic acid | Isovaleric acid |
| 5-Aminopentanoic acid | Malic acid |
| Anserine | Methylamine |
| L-Arginine | Nicotinic acid |
| Benzoic acid | Ornithine |
| Betaine | 2-Phenylpropionate |
| Butyric acid | Succinic acid |
| Choline | Taurine |
| Fumaric acid | Thymine |
| Homovanillic acid | Tyramine |
| p-Hydroxyphenylacetic acid | Uracil |
| myo-Inositol | Valeric acid |
| Isocaproic acid | |

**Saliva 25** — **25** — **Plasma 26**

### Specific plasma metabolites

| | |
|---|---|
| Acetone | 2-Hydroxyvaleric acid |
| L-Asparagine | Isobutyric acid |
| Capric acid | Isopropyl alcohol |
| Citric acid | D-Mannose |
| Creatinine | Methionine |
| Dimethylglycine | Methylsuccinic acid |
| Ethanol | 2-Oxovaleric acid |
| Glutamine | Phthalic acid |
| Histidine | Pyridoxine |
| 2-Hydroxybutyric acid | Serine |
| 2-Hydroxyisobutyrate | L-Threonine |
| (S)-3-Hydroxyisobutyric acid | L-Tryptophan |
| 3-Hydroxyisovaleric acid | Uridine |

**Fig 2. Venn diagram and list of specific and common metabolites found in mare saliva and plasma.**

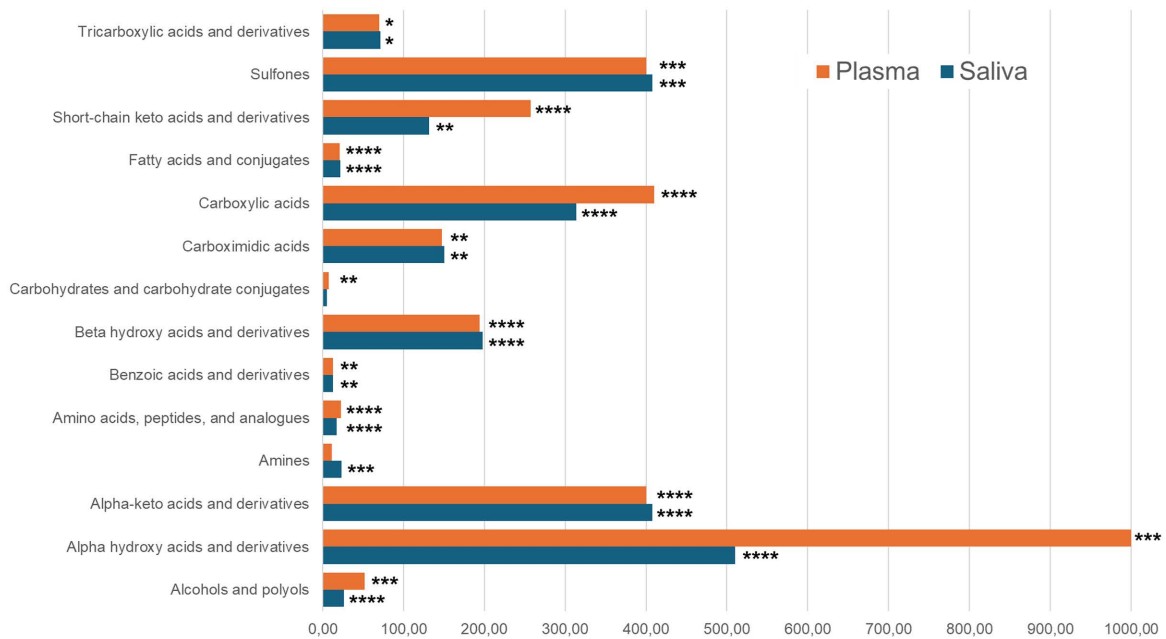

**Fig 3. Metabolite set enrichment analysis performed on the ¹H-NMR metabolome of mare saliva (blue) and plasma (orange).** The horizontal bars represent the enrichment ratio of the main chemical classes (metabolite sets) identified in comparison to a random distribution. p-values of each metabolite set are * $p < 0.05$, ** $p < 0.01$, *** $p < 0.005$, **** $p < 0.0001$.

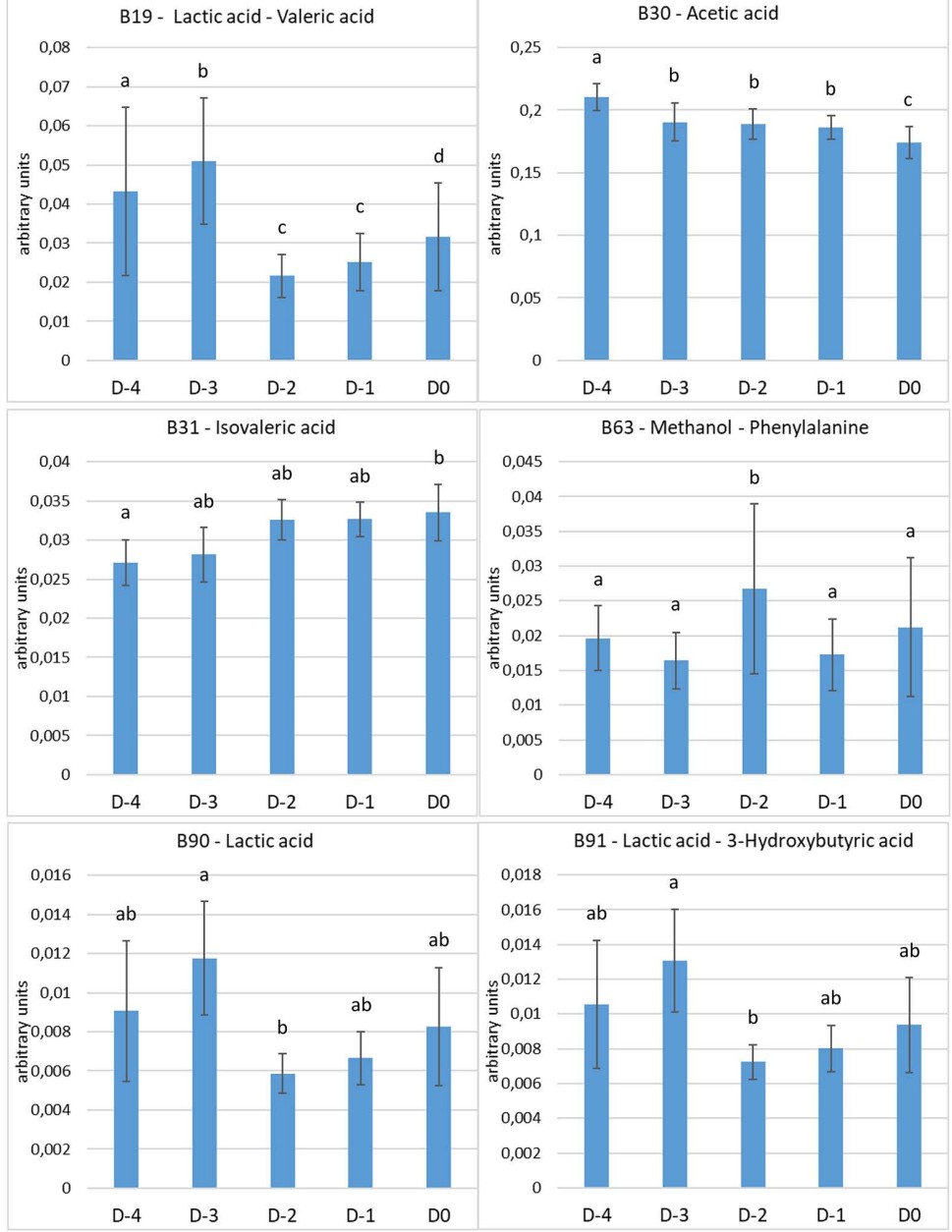

**Fig 4. Spectral intensities (arbitrary units, mean±sem) of the six metabolites or group of metabolites [chemical shift – name] from equine saliva showing significant differences between the days, normalized to the total area of all the spectral regions integrated.** a, b, c, d: values with different superscripts differ significantly (p<0.05).

individual variations. The intensities of the region containing Lactic acid and 3-Hydroxybutyric acid showed the same pattern. Because of the overlapping in bucket-91 of Lactic acid and 3-Hydroxybutyric acid, the most likely conclusion is that Lactic acid (B90-Lactic acid) is responsible for this change in bucket-91. This hypothesis is sustained by the fact that in bucket-19 (overlap of Lactic acid and Valeric acid) the same pattern of decrease from D-2 to parturition is observed,

leading to the conclusion that Lactic acid is responsible for this change. The salivary levels of all the other metabolites or groups of metabolites did not significantly change in our conditions during the five days before parturition.

In plasma, the levels of five metabolites or groups of metabolites with overlapping signals were significantly different between the days before parturition (Fig 5). The plasma level of D-Glucose significantly increased between three (D-3)

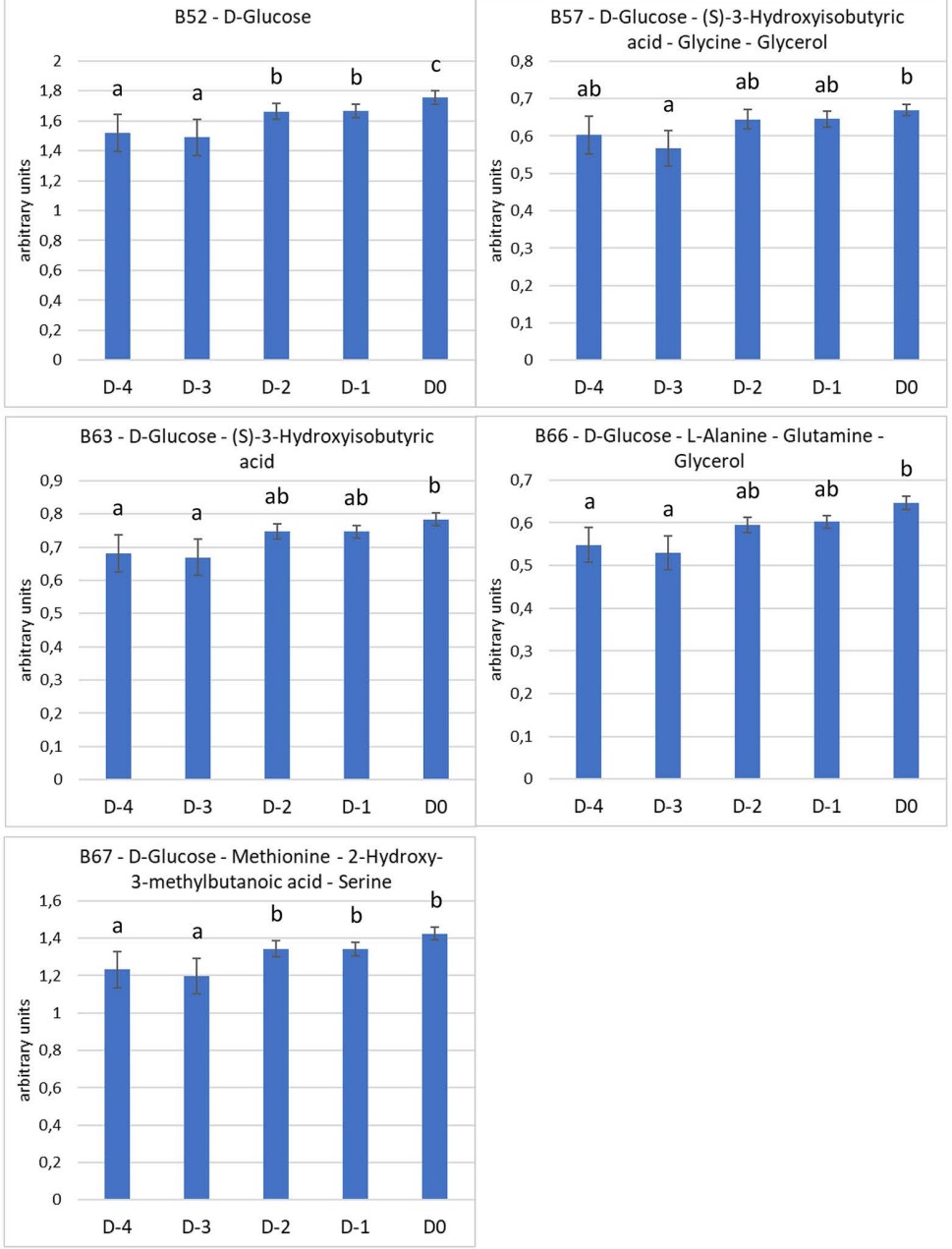

**Fig 5. Spectral intensities (arbitrary units, mean ± sem) of the five metabolites or group of metabolites [chemical shift – name] from equine plasma showing significant differences between the days, normalized to the total area of all the spectral regions integrated.** a, b, c: values with different superscripts differ significantly (p < 0.05).

and two (D-2) days before the day of foaling and significantly increased again on the day when parturition occurred (D0). The intensities (in arbitrary units) of the group of metabolites containing D-Glucose, (S)-3-Hydroxyisobutyric acid, Glycine and Glycerol significantly increased between three days before the day of foaling (D-3) and the day of parturition (DO). The intensities of the groups of metabolites containing D-Glucose and (S)-3-Hydroxyisobutyric acid, and D-Glucose, L-Alanine, Glutamine and Glycerol showed the same pattern. The intensities of the group of metabolites containing D-Glucose, Methionine, 2-Hydroxy-3-methylbutanoic acid and Serine significantly increased between three (D-3) and two (D-2) days before the day of parturition. These buckets (B57, B63, B66 and B67) contained Glucose and evolved the same way than B52 (containing only Glucose) during the days before foaling. It is likely that the level of Glucose impacted the behavior of these buckets. The plasma levels of all the other metabolites or groups of metabolites did not significantly change in our conditions during the five days before parturition.

## Discussion

In this prospective study, the use of $^1$H NMR to analyze multiple metabolites with high specificity and reproducibility provided opportunities to characterize, for the first time, the saliva metabolome during the antepartum period in the mare.

Modifications of saliva composition during the antepartum period have been shown in the mare for oxytocin and cortisol. Salivary oxytocin concentrations during the birth were higher than concentrations in saliva collected one day prior to parturition [16], but this increase at the time of foaling did not allow the prediction of the onset of foaling. Salivary cortisol concentrations increased during the last 2 days before foaling [17], but because of high individual variations, determination of salivary cortisol concentrations is not feasible for prediction of parturition [3]. Thus, the search for a salivary biomarker of the onset of parturition is still challenging.

Metabolome analysis of non-invasive fluids has been used to evaluate pregnancy stage or pregnancy outcome in mammals. The urinary metabolome revealed early pregnancy biomarkers and differential expression levels of urinary metabolites during pregnancy in sows [18], cows [19] and a cetacean (Yangtze finless porpoises, Neophocaena asiaeorientalis a.) [20]. During human pregnancy, the maternal urinary metabolome underwent significant changes and allowed to predict gestational age and time-to-delivery [21]. However, repeated urinary collection in animals can be challenging. Although saliva collection is easier, salivary metabolome analysis during pregnancy are scarce. The saliva metabolome analysis in early pregnant and non-pregnant sows revealed early pregnancy biomarkers [22]. In a previous study, we analyzed saliva metabolome during seasonal anoestrus, the oestrus cycle and early gestation in the mare, and identified metabolites whose saliva levels vary significantly between the targeted physiological stages [15]. To our knowledge, no other data on saliva metabolome during gestation and parturition are available.

In the present study, we identified 50 metabolites in saliva and 51 in plasma, 25 of them were present in both biofluids, including amino acids such as L-Alanine, Glycine, Leucine, Isoleucine, L-Tyrosine and L-Valine, organic acids such as Formic acid, Hippuric acid, Acetic acid and Propionic acid, components of energy metabolism pathways such as Creatine, Phosphocreatine, D-Glucose, Lactic acid and Pyruvic acid. Common metabolites support the use of saliva as a non-invasive proxy of plasma for diagnostic purposes.

Among the 50 metabolites identified in saliva, 25 of them were present only in saliva, including amino acids, peptides and derivatives such as L-Arginine, Anserine, Betaine, Ornithine and Taurine, nucleic bases such as Thymine and Uracile, organic acids such as Benzoic acid, Butyric acid, Fumaric acid, Isocaproic acid, Isovaleric acid, Malic acid, Nicotinic acid, Succinic acid and Valeric acid. Saliva is rich in molecular diversity and numerous chemical classes are secreted in it. Its non-invasive sampling and its significant molecular richness make it an interesting fluid. This observation emphasis the fact that saliva could be a biofluid of choice when looking for biomarkers.

In the saliva of antepartum mares, the levels of Acetic acid, Isovaleric acid, Lactic acid and three groups of metabolites showed significant modifications in the five days prior to foaling. Acetic acid levels were significantly lower on the day of foaling compared to the other days, Isovaleric acid levels were significantly higher on the day of foaling compared to

four days before, Lactic acid levels significantly decreased two days before the day of foaling. Moreover, the levels of the groups containing Lactic acid and Valeric acid, Methanol and Phenylalanine, and Lactic acid and 3-Hydroxybutyric acid showed significant modifications in the antepartum period, but $^1$H NMR analysis did not allow us to assign these buckets to a single metabolite because the signals of these metabolites overlapped. However, it is fair to conclude that the groups of Lactic acid and 3-Hydroxybutyric acid and Lactic acid and Valeric acid were impacted by the Lactic acid fluctuations during these 5 days since they have the same evolution as Lactic acid alone.

To our knowledge, only two studies have been published about salivary concentrations of Lactic acid in the antepartum period in mammals. In dairy cows, salivary concentrations of Lactic acid are significantly lower within the 12 first hours after calving than 13 days before calving [23]. However, no data are available on the evolution of salivary levels of Lactic acid between 13 days before calving and the calving day. In sows, salivary concentrations of Lactic acid significantly increased between early gestation (30 days of gestation) and late gestation (90 days of gestation, 24 days before farrowing), and were not significantly different between late gestation and the first 24 hours after farrowing [24]. However, no data are available on the evolution of salivary levels of Lactic acid between 24 days before farrowing and the farrowing day. Thus, our study is the first analysis of salivary levels of Lactic acid during the last days before parturition in mammals.

Plasma concentrations of Lactic acid have been analyzed in the antepartum period in cows, sows, and women. In cows, daily blood sampling before parturition showed Lactic acid increase from one day before calving to calving [25]. Sows at onset of farrowing had increased plasma concentrations of Lactic acid, when compared with sows on day 107 of gestation [26]. In late gestation and during farrowing, Lactic acid is released from the sow uterus to the blood, reflecting that the uterus is working and part of the energy metabolism is anaerobic [27]. Moreover, in women, Lactic acid concentrations increase in labor, a period of intense physical exertion and anaerobic respiration [28]. In our study, the variations of plasma levels of Lactic acid before foaling could not be analyzed because Lactic acid signal in plasma overlapped with Threonine and 3-Hydroxybutyric acid signals. However, the variations of Lactic acid levels were analyzed in saliva, and a non-significant increase of saliva levels of Lactic acid was observed during the last two days before foaling. This increase could be related to the start of uterine contractions, and the release of Lactic acid from the mare uterus to the blood and its passage to the saliva. Unfortunately, in our study, we could not analyze the concomitant variations of saliva and plasma levels. Plasma levels of Lactic acid should be analyzed with another technique to ascertain whether or not they significantly change in the prepartum period. In equine species, saliva and plasma concentrations of Lactic acid were analyzed in endurance horses during a standardized exercise test, and no correlation was observed between saliva and plasma Lactic acid concentrations [29].

In our study, Acetic acid was found in saliva and plasma of the antepartum mares. Saliva levels were significantly lower on the day of foaling compared to the other days, whereas plasma levels did not significantly change. To our knowledge, only two publications investigated salivary levels of Acetic acid during gestation. Acetic acid was detected in the saliva of pregnant women in the 3rd trimester of pregnancy, but no data were available about saliva levels when parturition approach [30,31]. Thus, our study is the first analysis of Acetic acid levels in the saliva in the antepartum period in mammals. Short-chain fatty acids, such as Acetic acid, can be utilized as energy source in herbivorous animal species and Acetic acid is used principally by muscle [32]. Thus, the lower saliva level of Acetic acid may be related to the energy expenditure by the start of uterine contractions. However, the lack of a concomitant decrease in Acetic acid levels in saliva and plasma remains unexplained. The plasma levels of Acetic acid during gestation was only analyzed in sows, and decreased plasma concentrations were observed at onset of farrowing when compared with sows on day 107 of gestation [26].

In our study, Isovaleric acid was detected only in saliva, and levels were significantly higher on the day of foaling compared to four days before. There are limited publications in the literature that investigate salivary levels of Isovaleric acid during gestation. In the saliva of pregnant women in the 3rd trimester of pregnancy, Isovaleric acid was detected, but no data were available about saliva levels when parturition approach [30,31]. Thus, our report of variations of Isovaleric acid levels in the saliva from antepartum mare is unique among domestic mammals.

In the present study, a concomitant analysis of plasma samples was performed, to analyze simultaneous modifications of the metabolome in saliva and plasma. The plasma metabolomic profile was previously characterized in five healthy pregnant mares between days 285 and 290 of gestation using NMR spectroscopy [33] and nine healthy pregnant mares during peri-parturition using gas chromatography–mass spectrometry (GC-MS) [34] with 54 and 95 metabolites identified respectively. In our study, the composition of the plasma metabolome is consistent with these previous reports.

In our conditions, the plasma levels of D-glucose and four groups of metabolites containing D-glucose showed significant modifications in the five days prior to foaling. The plasma level of D-glucose significantly increased from three days before the day of foaling to the day of foaling. D-glucose was also found in concomitant saliva samples, but no significant variations of the saliva levels of D-glucose was detected in the antepartum period. The plasma level of the four groups of metabolites containing D-glucose showed significantly higher levels on the day of foaling than on another day. However, the $^1$H NMR analysis did not allow us to assign these buckets to a single metabolite because the signals of these metabolites overlapped. Their levels should be analyzed with another technique to ascertain whether or not they significantly change in the prepartum period. However, the evolution of these groups of metabolites containing the D-Glucose is similar to those of D-Glucose alone, suggesting that only D-Glucose evolves during the five days before parturition.

In a previous study, plasma metabolomic analysis in peri-parturient mares identified metabolites that exhibited significant changes in the last 4 days before foaling, but D-glucose was not detected [34]. Moreover, this study showed that Glycerol-3-phosphate could be a potential biomarker for predicting parturition, because its plasma level sharply increased 3 days before parturition. However, this metabolite was not detected in our conditions. The discrepancy between this study and our study could be related to the metabolomic techniques used: GC-MS versus NMR spectroscopy respectively.

The antepartum increase of Glucose level observed in our study was not related to feeding modifications since mares lived in the pasture from April to foaling in May and June. It may be related to the antepartum increase in plasma cortisol concentration that promotes gluconeogenesis [17], or to changes in carbohydrate metabolism and pancreatic beta-cell function during pregnancy that may contribute to progressive development of insulin resistance during late gestation [35]. This increase in Glucose concentration may be linked to the increasing demands of the fetus and the energy requirements of maternal tissues. Glucose has been shown to be the key energy metabolite for oxidative metabolism of gravid uterus in the sow [27]. Thus, the increase in plasma level of Glucose between 3 days before the day of foaling and 2 days before the day of foaling may be responsible for the concomitant significant decrease of saliva level of Lactic acid. High level of Lactic acid may reflect that insufficient Glucose was available to fuel the start of uterine contractions, and low levels of Lactic acid may reflect sufficient supply of glucogenic energy. Similar variations of Glucose level were obtained from Heavy Draft Horse and Standardbred mares, with Glucose level significantly higher at parturition than 7–10 days before parturition [36,37].

In dairy cows, the plasma concentration of Glucose significantly increased from 7 days before calving to the calving [38]. In the pregnant cow, a prepartum increase in blood Glucose concentration one day before parturition was observed and proposed as an indicator to detect calving [39]. Moreover, the development of a wireless sensor affixed at the base of the ventral tail of the cow for monitoring tissue Glucose concentration allowed calving prediction within 12h [40]. A continuous Glucose monitoring system based on interstitial Glucose measurement has been tested on stallions and foals, showing a good agreement between interstitial and blood Glucose concentrations [41,42]. The development of non-invasive Glucose monitoring sensors [43], and the adaptation of these sensors to the mare, could allow real-time, painless detection of a prepartum increase in plasma Glucose level, that could be a non-invasive biomarker of the onset of foaling.

In conclusion, significant changes in the salivary metabolome have been shown in the antepartum period in the mare, in particular for Acetic acid, Isovaleric acid and Lactic acid. Thus, saliva is a relevant biofluid for monitoring parturition. However, we observed only minor changes in the levels of these metabolites, without any significant threshold that would allow the prediction the onset of foaling. Further studies are therefore necessary to identify non-invasive salivary biomarkers for the prediction of parturition. In addition, significant modifications of the plasma level of Glucose have been

observed before foaling. The development of non-invasive Glucose monitoring sensors could allow the development of non-invasive detection method for the prediction of foaling.

## Supporting information

**S1 Table. Identifications of the metabolites with their chemical shifts in ppm, their multiplicities and their corresponding identifiers according to the Metabolomics Standards Initiative (MSI).**
(XLSX)

**S2 Table. List of metabolites in each chemical class from saliva and plasma.** Compounds that are common to the two biofluids are in italic.
(DOCX)

## Acknowledgments

We would like to thank Philippe Barrière, Thierry Blard, Antoine Crampon and Yvan Gaude for their technical contribution to this work, and Gaëlle Lefort for her help in the statistical analysis. We would like to thank the "Plateforme de Métabolomique et d'Analyses Chimiques", US61 ASB of Tours University.

## Author contributions

**Conceptualization:** Lydie Nadal-Desbarats, Amandine Gesbert, Fabrice Reigner, Ghylène Goudet.

**Data curation:** Lydie Nadal-Desbarats, Ghylène Goudet.

**Formal analysis:** Camille Dupuy, Frédéric Montigny.

**Funding acquisition:** Lydie Nadal-Desbarats, Fabrice Reigner, Ghylène Goudet.

**Investigation:** Lydie Nadal-Desbarats, Camille Dupuy, Frédéric Montigny, Priscila Silvana Bertevello, Cécile Douet, Amandine Gesbert, Fabrice Reigner, Ghylène Goudet.

**Methodology:** Lydie Nadal-Desbarats, Camille Dupuy, Frédéric Montigny, Priscila Silvana Bertevello, Cécile Douet, Amandine Gesbert, Fabrice Reigner, Ghylène Goudet.

**Project administration:** Lydie Nadal-Desbarats, Fabrice Reigner, Ghylène Goudet.

**Resources:** Cécile Douet, Amandine Gesbert, Fabrice Reigner, Ghylène Goudet.

**Supervision:** Lydie Nadal-Desbarats, Amandine Gesbert, Fabrice Reigner, Ghylène Goudet.

**Validation:** Lydie Nadal-Desbarats, Ghylène Goudet.

**Visualization:** Lydie Nadal-Desbarats, Camille Dupuy, Frédéric Montigny, Priscila Silvana Bertevello, Ghylène Goudet.

**Writing – original draft:** Lydie Nadal-Desbarats, Ghylène Goudet.

**Writing – review & editing:** Lydie Nadal-Desbarats, Camille Dupuy, Frédéric Montigny, Priscila Silvana Bertevello, Cécile Douet, Amandine Gesbert, Fabrice Reigner, Ghylène Goudet.

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
