## [Decision Letter · Decision Letter 0]

27 Jan 2026

Dear Dr. Goudet,

Thank you for submitting your manuscript to PLOS ONE. After careful consideration, we feel that it has merit but does not fully meet PLOS ONE’s publication criteria as it currently stands. Therefore, we invite you to submit a revised version of the manuscript that addresses the points raised during the review process.

We look forward to receiving your revised manuscript.

Kind regards,

Juan J Loor

Academic Editor

PLOS One

Journal Requirements:

2. To comply with PLOS One submissions requirements, in your Methods section, please provide additional information regarding the experiments involving animals and ensure you have included details on (1) methods of sacrifice, (2) methods of anesthesia and/or analgesia, and (3) efforts to alleviate suffering.

3. We note that this submission includes NMR spectroscopy data. We would recommend that you include the following information in your methods section or as Supporting Information files:

1) The make/source of the NMR instrument used in your study, as well as the magnetic field strength. For each individual experiment, please also list: the nucleus being measured; the sample concentration; the solvent in which the sample is dissolved and if solvent signal suppression was used; the reference standard and the temperature.

2) A list of the chemical shifts for all compounds characterised by NMR spectroscopy, specifying, where relevant: the chemical shift (δ), the multiplicity and the coupling constants (in Hz), for the appropriate nuclei used for assignment.

3)The full integrated NMR spectrum, clearly labelled with the compound name and chemical structure.

We also strongly encourage authors to provide primary NMR data files, in particular for new compounds which have not been characterised in the existing literature. Authors should provide the acquisition data, FID files and processing parameters for each experiment, clearly labelled with the compound name and identifier, as well as a structure file for each provided dataset. See our list of recommended repositories here: https://journals.plos.org/plosone/s/recommended-repositories

4. Please expand the acronym “INRAE” (as indicated in your financial disclosure) so that it states the name of your funders in full.

“This work was supported by INRAE and Institut Français du Cheval et de l’Equitation (IFCE).”

Reviewers' comments:

Reviewer's Responses to Questions

**Comments to the Author**

1. Is the manuscript technically sound, and do the data support the conclusions?

Reviewer #1: Yes

2. Has the statistical analysis been performed appropriately and rigorously?

Reviewer #1: Yes

3. Have the authors made all data underlying the findings in their manuscript fully available?

Reviewer #1: Yes

4. Is the manuscript presented in an intelligible fashion and written in standard English?

Reviewer #1: Yes

Reviewer #1: The study describes potential biomarkers measurable in the saliva of pregnant mares to determine parturition. This approach is particularly valuable because saliva can be collected non-invasively by non-veterinary personnel. The study is well designed, and only minor revisions are required.

Abstract, line 29: Replace “level” with “concentration” and apply this change consistently throughout the manuscript whenever quantitative data are reported.

Discussion, line 370:

Original: “However, the lack of concomitant decrease of acetic acid levels in saliva and plasma remains questioning.”

Suggested correction:

“However, the lack of a concomitant decrease in acetic acid concentrations in saliva and plasma remains unclear (or unexplained).”

Lines 155–156: Insert “samples” for clarity:

“The 60 saliva samples were prepared by protein precipitation using cold methanol.”

**Do you want your identity to be public for this peer review?** For information about this choice, including consent withdrawal, please see our Privacy Policy

Reviewer #1: No

---

## [Author Response · Author response to Decision Letter 1]

6 Feb 2026

Journal Requirements:

1. We have modified the manuscript so that it meets PLOS ONE’s style requirements, the modifications have been highlighted.

2. Information regarding the experiments involving animals and details on methods of sacrifice, anesthesia, analgesia, and efforts to alleviate suffering have been included and highlighted in the Methods section lines 115 to 151, and in the Ethics statement section lines 452 to 464.

3. We have included the requested information about NMR spectroscopy data:

1) In the methods section (line 172 to 179 in the revised manuscript), we have already mentioned the source of the NMR instrument, the magnetic field strength, the nucleus being measured, the solvent, the temperature and the pulse sequence to supress water signal: “1H-NMR spectra were acquired at 298 K on an AVANCE III HD 600 MHz system (Bruker Biospin, Karlsruhe, Germany) equipped with a 5 mm TCI cryoprobe with z-gradient. 1H-NMR spectra were recorded with a “noesypr1d” pulse sequence with a time domain of 64 K data points, a sweep width of 12 ppm, 64 scans, an acquisition time of 4.56s and a relaxation delay of 20s. A NUS (50%) TOCSY sequence was used on the quality controls for the metabolite identifications of the biological samples. The 1H-NMR spectra were phase corrected and imported in NMRProcFlow (DOI: 10.1007/s11306-017-1178-y) for baseline correction and bucketing. Regions with a coefficient of variation across the quality control inferior to 30% were kept for the analysis.”

Lines 169 to 171, the volume of solvent and the standard concentration are already mentioned: “Before NMR analysis, 210 µL of deuterated phosphate buffer (pH = 7.4; containing 3-trimethylsilylpropionic acid (TSP) at 152 µM final concentration) were added to the dried residues and transferred to 3 mm NMR tubes.”

2) The S1-Table resumes, for identified compounds in our biological samples, their chemical shifts in ppm, their multiplicities and their corresponding identifiers according to the Metabolomics Standards Initiative (MSI).

3) Full annotated representative 1H-Nuclear Magnetic Resonance spectroscopy spectra of mare saliva and plasma are in Figure 1 (A) and (B). Concerning the chemical structure, the HMDB, PubChem identifiers are mentioned in S1-Table and their SMILES identifiers related to their structure are also resumed in the S1-Table.

The chemical structures were intentionally omitted from the Figure 1 to prevent visual overcrowding.

The NMR data were already deposited on Zenodo Open repository. The data can be uploaded using the following link: https://doi.org/10.5281/zenodo.17589754. Line 201 to 202 of the manuscript mentioned: “The experimental data have been deposited with the DOI 10.5281/zenodo.17589754 in the Zenodo repository (https://zenodo.org).”

4. We have expanded the acronym INRAE to ‘Institut National de Recherche pour l’Agriculture, l’Alimentation et l’Environnement’, and we have highlighted the modifications.

5. We have stated the financial disclosure and the role of the funders and highlighted these changes lines 447 to 450:

Financial disclosure

This work was supported by Institut National de Recherche pour l’Agriculture, l’Alimentation et l’Environnement (INRAE) and Institut Français du Cheval et de l’Equitation (IFCE). The funders had no role in study design, data collection and analysis, decision to publish, or preparation of the manuscript.

6. The reviewer comments did not include a recommendation to cite specific previously published works.

7. We have reviewed our reference list, it is complete and correct.

Response to Reviewers’ comments:

5. Review comments to the author:

Abstract, line 29: Replace “level” with “concentration” throughout the manuscript

We thank the reviewer to raise this question. However, the authors deliberately adopted the term “level” rather than “concentration”, because although the spectral intensities are expressed relative to a reference standard, they are not corrected for the number of protons contributing to each signal cluster and therefore cannot be interpreted as absolute concentrations. The data presented correspond to relative signal intensities; accordingly, the use of the term “level” is retained.

Discussion, line 370 (373 in the revised manuscript): the sentence has been modified and highlighted.

Lines 155-156 (159 in the revised manuscript): the sentence has been modified and highlighted.

We have reviewed the figures so that they meet the technical requirements.

---

## [Editor Report · Decision Letter 1]

22 Feb 2026

Saliva and plasma metabolome analysis during the five days before foaling in the mare

PONE-D-25-61361R1

Dear Dr. Goudet,

We’re pleased to inform you that your manuscript has been judged scientifically suitable for publication and will be formally accepted for publication once it meets all outstanding technical requirements.

Kind regards,

Juan J Loor

Academic Editor

PLOS One
---

## [Editor Report · Acceptance letter]

PONE-D-25-61361R1

PLOS One

Dear Dr. Goudet,

I'm pleased to inform you that your manuscript has been deemed suitable for publication in PLOS One. Congratulations! Your manuscript is now being handed over to our production team.

Kind regards,

on behalf of

Dr. Juan J Loor

Academic Editor

PLOS One